

# Relation between lymphocyte to monocyte ratio and survival in patients with hypertrophic cardiomyopathy: a retrospective cohort study

Zhonglan Chen[1,*], Ziqiong Wang[2,*], Youping Li[3], Xiaoping Chen[2] and Sen He[2]

[1] West China Hospital Cardiology department/West China School of Nursing, Chinese Evidence-Based Medicine Centre, Cochrane China Center, Chengdu, China
[2] Department of Cardiology, West China Hospital of Sichuan University, Chengdu, China
[3] Chinese Evidence-Based Medicine Centre, Cochrane China Center, West China Hospital Sichuan University, Chengdu, China
[*] These authors contributed equally to this work.

## ABSTRACT

**Background**. The lymphocyte-to-monocyte ratio (LMR) has been proposed as a novel prognostic factor in malignancies and cardiovascular diseases. Our study aimed to ascertain whether LMR is a useful biomarker in discriminating the hypertrophic cardiomyopathy (HCM) patients at higher risk of all-cause mortality.

**Methods**. This retrospective study consisted of 354 adult HCM patients. Cox's proportional hazards regression models were used to analyze the association between LMR and all-cause mortality. Smooth curve fitting was conducted to explore the linear relationship between LMR and all-cause mortality.

**Results**. During the follow-up, 44 patients reached the study endpoint. The all-cause mortality rate was 7.3 per 100 person-years in the first tertile and decreased across the three tertiles of LMR. With the first tertile as reference, adjusted hazard ratios (HR) for all-cause mortality were 0.43 for the second tertile (95% CI [0.20–0.91], $p = 0.027$) and 0.39 for the third tertile (95% CI [0.17–0.90], $p = 0.028$), respectively. Smooth curve fitting exhibited a nonlinear relationship between LMR values and all-cause mortality. For LMR < 6.5, per SD increase resulted in a significantly decreased risk of all-cause mortality by 62% (HR: 0.38, 95% CI [0.21–0.68]). For LMR ≥ 6.5, the all-cause mortality risk did not progressively increase. Stratified and subgroup analyses revealed similar results to the main analyses,andE-value analysis suggested robustness to unmeasured confounding.

**Conclusions**. The study demonstrated that LMR was an independent predictor of all-cause mortality in HCM patients, and LMR may be useful for identifying HCM patients at high mortality risk.

Corresponding author
Sen He, hesensubmit@163.com

## INTRODUCTION

Hypertrophic cardiomyopathy (HCM) is viewed as the most common inherited heart disease with an estimated prevalence of 0.2% of the general population (*Antunes & Scudeler, 2020*). The disease exhibits remarkable heterogeneity in clinical manifestations and natural history. Patients with HCM can be asymptomatic or oligosymptomatic or suffered from progressive heart failure, thromboembolic events, malignant arrhythmic events and unpredicted cardiac death (*Makavos et al., 2019*). Therefore, the discovery of novel inflammatory biomarkers has become meaningful so as to provide information of prognosis and guide clinical management in the disease.

Historical studies have suggested the existence of low grade systemic and local inflammation in patients with HCM (*Fang et al., 2017*; *Becker, Owens & Sadayappan, 2020*). Tissue-level inflammation is documented by immunohistological evidence of invading inflammatory lymphocytes and monocytes in the cardiac tissue (*Lamke et al., 2003*; *Kuusisto et al., 2012*). Importantly, there was a significant correlation between the degree of inflammatory cell infiltration and myocardial fibrosis in these diseased hearts (*Westermann, 2012*). Besides, a series of systemic inflammatory biomarkers, including neutrophil to lymphocyte ratio, monocyte to high-density lipoprotein cholesterol ratio and high-sensitivity C reactive protein, all showed promising results in risk stratification in HCM (*Ozyilmaz et al., 2017*; *Ekizler et al., 2019*; *Zhu et al., 2017*). Patients with increased levels of those biomarkers tend to be endangered by poor prognosis. Lymphocytes and monocytes are important immune cells in the inflammatory process.

The ratio of lymphocyte to monocyte, namely, LMR, has been reported to be significantly associated with the prognosis of various malignancies (*Nishijima et al., 2015*) and cardiovascular disorders (*Wang et al., 2017*; *Kiris et al., 2017*; *Ren et al., 2017*; *Park et al., 2018*). In patients with ST-elevated myocardial infarction after primary percutaneous coronary intervention, both in hospital and long-term adverse cardiac and cerebrovascular events, such as nonfatal myocardial infarction, stroke, as well as cardiovascular mortality decreased with increasing LMR (*Wang et al., 2017*; *Kiris et al., 2017*). In patients with acute heat failure, a lower value of LMR was associated with a higher mortality risk within 6 months (*Ren et al., 2017*). However, up to now, there is no data in HCM regarding LMR as a prognostic marker. As both lymphocytes and monocytes played important role in myocardial inflammation and fibrosis in HCM, we aimed to investigate whether LMR is a useful biomarker in discriminating the HCM patients at higher risk of all-cause mortality.

## METHODS

### Study population

This is a retrospective study where all HCM patients greater than 18 years of age who have been treated at Sichuan University West China Hospital, which is a tertiary referral center in Southwest of China were consecutively enrolled, forming the inception cohort of 508 patients. The enrollment started at Dec 2008 and ended at May 2016. The diagnosis of HCM was based on the presence of increased left ventricular wall thickness ($\geq$15 mm), identified by echocardiography, cardiac magnetic resonance or computed tomography, that was not

solely explained by abnormal loading conditions (*Zamorano et al., 2014*). Patients with other inherited metabolic diseases or syndromic causes of HCM (5 cardiac amyloidosis, 2 restrictive cardiomyopathy, 1 dilated cardiomyopathy and 1 myocarditis) were excluded from the study. For the present study, other exclusion criteria were as follows: (1) patients with comorbidities which would affect the white blood cell count (WBCC) were excluded ($n = 91$), such as infections, hematological system diseases and taking corticosteroids; (2) patients with age under 18 ($n = 5$) and patients with incomplete data ($n = 26$) were further excluded from the study; (3)We also excluded patients who were lost to follow-up after the first evaluation ($n = 23$). Finally, the present study included 354 adult HCM patients for the final analysis.

This study was approved by the Biomedical Research Ethics Committee, West China Hospital of Sichuan University (approval number: 2019-1147). This study was performed in keeping with the criteria set by Declaration of Helsinki. Due to the retrospective nature of the study, informed consent was waived. Other detailed information has been reported in the recently published studies (*Wang et al., 2020*).

## Baseline clinical evaluation

Complete blood count (CBC) was measured by a Sysmex XN-9000 analyzer (Sysmex Corporation, Kobe, Japan) at the time of hospital admission, and the laboratory parameters of CBC included number of WBCC and percentage of lymphocyte and monocyte, and so on. The normal range for WBCC was $4-10^\star 10^9$/L in this system. The LMR was calculated based on the results of CBC. All patients underwent 2D transthoracic echocardiography examinations by standard techniques at baseline, and the data and other detailed information of baseline characteristics were collected from electronic medical records.

## Study endpoint

The endpoint of the study was all-cause mortality. Follow-up was carried out by medical records review or telephone contacting with patients or their family members. All patients were followed until the date of death or the end of follow-up at Feb 2020, whichever came first.

## Statistical analysis

To quantify in a simple form the association between LMR and all-cause mortality, the patients were divided into three groups according to baseline LMR: tertile 1 (<3.8), tertile 2 (3.8–5.4) and tertile 3 (≥5.5) (*Wang et al., 2017*; *Kiris et al., 2017*; *Mabikwa et al., 2017*). For each group, continuous variables were presented as mean ± standard deviation (SD) and median with interquartile range (IQR) where appropriate, and categorical variables as number (percentage). The baseline characteristics among the three groups were analyzed by using the analysis of variance or Kruskal-Wallis tests for continuous variables, and the chi-square or Fisher exact tests for categorical variables.

The cumulative mortality in each group was calculated by Kaplan–Meier method, and the log-rank test was used for comparison. Cox proportional hazard regression analysis was used to assess the role of LMR as an independent predictor of mortality.
Variables for inclusion were carefully chosen to ensure parsimony of the multivariate models. Age and gender were forced into the multivariate models, and baseline variables that showed a univariate relationship with mortality ($p < 0.05$) or that were considered clinically relevant (*Liu et al., 2017*) were included for adjustment in multivariate models. Additionally, we explored the association between LMR and all-cause mortality by smooth curve fitting after adjustment for potential confounders, and a two-piecewise Cox model was also applied to examine the threshold effect of LMR on mortality. Additionally, a time-dependent receiver operating characteristic (ROC) curve was generated to evaluate the accuracy of LMR in the discrimination of mortality at different prediction times (*Kamarudin, Cox & Kolamunnage-Dona, 2017*). A generally accepted approach suggests that the area under curves (AUCs) of less than 0.60 reflects poor discrimination; 0.60 to 0.75, possibly helpful discrimination; and more than 0.75, clearly useful discrimination.

The robustness of these findings was assessed in multiple sensitivity analyses. First, stratified analysis assessed the consistency of association between LMR and mortality in different groups. Second, we explored the potential for unmeasured confounding between LMR and all-cause mortality by calculating E-values (*Haneuse, VanderWeele & Arterburn, 2019*). The *E*-value quantifies the required magnitude of an unmeasured confounder that could negate the observed association between LMR and all-cause mortality, and the *E*-value was calculated on the web calculator (https://www.evalue-calculator.com/). Third, we also assessed the relation between LMR and mortality in the subgroup with a normal WBCC range as another sensitivity analysis.

All analyses were performed with R version 3.6.3 including the "compareGroups", "rms", "survminer", "tidyverse", "survival" "timeROC" and "base" packages (http://www.R-project.org). All tests were two sided, and *p* values <0.05 were considered statistically significant.

## RESULTS

### Baseline characteristics

Table 1 shows the baseline characteristics of the study cohort. The average age was 56 (IQR: 44.0–66.0) and male patients accounted for 55.1%. The LMR distribution ranged from 0.2–17.4. There were 115 patients in title 1, 116 patients in tertile 2 and 123 patients in tertile 3. The lymphocytes increased significantly across the LMR tertiles. On the contrary, WBCC, neutrophils and monocytes decreased significantly across the LMR tertiles. Besides, patients with higher LMR values tended to be younger and have significantly lower left atria diameter, as well as significantly higher hemoglobin concentration and left ventricular ejection fraction. Other clinical features did not show significant difference among the three groups.

### Survival analysis

During the follow-up, 44 patients reached the endpoint of all-cause mortality. The specific causes of deaths were as follows: 7 sudden cardiac deaths, 12 heart failure-related deaths, 9 stroke-related deaths, 2 HCM-related postoperative deaths, 1 other cardiac death and

**Table 1  Baseline characteristics of study cohort.**

| Variable | All patients (n = 354) | LMR, tertile 1 (n = 115) | LMR, tertile 2 (n = 116) | LMR, tertile 3 (n = 123) | p value |
|---|---|---|---|---|---|
| Age (years) | 56.0 (44.0–66.0) | 58.0 (44.5–67.0) | 58.5 (47.8–67.0) | 51.0 (39.5–63.5) | 0.012* |
| Gender, male | 195 (55.1%) | 64 (55.7%) | 66 (56.9%) | 65 (52.8%) | 0.811 |
| Family history of HCM | 35 (9.89%) | 9 (7.83%) | 10 (8.62%) | 16 (13.0%) | 0.350 |
| Family history of SCD | 16 (4.52%) | 8 (6.96%) | 6 (5.17%) | 2 (1.63%) | 0.130 |
| Symptoms | | | | | |
| Chest pain | 201 (56.8%) | 61 (53.0%) | 72 (62.1%) | 68 (55.3%) | 0.352 |
| Palpitation | 149 (42.1%) | 45 (39.1%) | 46 (39.7%) | 58 (47.2%) | 0.370 |
| Syncope/pre-syncope | 119 (33.6%) | 35 (30.4%) | 47 (40.5%) | 37 (30.1%) | 0.158 |
| Dyspnea | 201 (56.8%) | 74 (64.3%) | 63 (54.3%) | 64 (52.0%) | 0.129 |
| Medical history | | | | | |
| Atrial fibrillation | 60 (16.9%) | 21 (18.3%) | 19 (16.4%) | 20 (16.3%) | 0.901 |
| Hypertension | 112 (31.6%) | 42 (36.5%) | 38 (32.8%) | 32 (26.0%) | 0.209 |
| Diabetes | 30 (8.47%) | 13 (11.3%) | 8 (6.90%) | 9 (7.32%) | 0.412 |
| Vascular disease | 25 (7.06%) | 10 (8.70%) | 8 (6.90%) | 7 (5.69%) | 0.662 |
| Prior TE | 15 (4.24%) | 7 (6.09%) | 6 (5.17%) | 2 (1.63%) | 0.159 |
| Therapy | | | | | |
| Aspirin | 66 (18.6%) | 19 (16.5%) | 28 (24.1%) | 19 (15.4%) | 0.176 |
| Clopidogrel | 17 (4.80%) | 9 (7.83%) | 4 (3.45%) | 4 (3.25%) | 0.182 |
| β-blocker | 273 (77.1%) | 85 (73.9%) | 89 (76.7%) | 99 (80.5%) | 0.479 |
| ACE inhibitor or ARB | 65 (18.4%) | 20 (17.4%) | 21 (18.1%) | 24 (19.5%) | 0.911 |
| Pacemaker | 16 (4.52%) | 8 (6.96%) | 4 (3.45%) | 4 (3.25%) | 0.628 |
| ICD | 28 (7.91%) | 10 (8.70%) | 9 (7.76%) | 9 (7.32%) | |
| Alcohol septal ablation | 33 (9.32%) | 8 (6.96%) | 12 (10.3%) | 13 (10.6%) | 0.694 |
| Septal myectomy | 6 (1.69%) | 1 (0.87%) | 3 (2.59%) | 2 (1.63%) | |
| Hematological | | | | | |
| Hemoglobin (g/L) | 139 (126–151) | 137 (120–150) | 136 (124–150) | 142 (132–154) | 0.022* |
| WBCC ($10^9$/L) | 6.25 (5.15–7.49) | 7.00 (5.50–9.02) | 6.19 (5.15–7.09) | 5.87 (5.05–6.86) | <0.001* |
| Neutrophil ($10^9$/L) | 3.79 (3.00–4.92) | 4.87 (3.45–6.58) | 3.78 (3.04–4.68) | 3.44 (2.64–3.93) | <0.001* |
| Lymphocyte ($10^9$/L) | 1.65 (1.32–2.02) | 1.35 (0.98–1.61) | 1.63 (1.38–1.94) | 1.97 (1.73–2.34) | <0.001* |
| Monocyte ($10^9$/L) | 0.35 (0.27–0.46) | 0.47 (0.39–0.60) | 0.35 (0.30–0.44) | 0.28 (0.23–0.33) | <0.001* |
| LMR | 5.02 ±2.32 | 2.76 ±0.82 | 4.61 ±0.48 | 7.53 ±1.85 | <0.001* |
| Lipid profiles | | | | | |
| LDL-C (mmol/L) | 2.42 (1.84–2.90) | 2.38 (1.84–2.92) | 2.33 (1.81–2.84) | 2.55 (1.96–2.95) | 0.489 |
| HDL-C (mmol/L) | 1.27 (1.02–1.53) | 1.30 (1.02–1.54) | 1.21 (1.00–1.48) | 1.28 (1.04–1.52) | 0.325 |
| TG (mmol/L) | 1.25 (0.94–1.92) | 1.25 (0.96–1.68) | 1.27 (0.96–2.10) | 1.20 (0.90–1.99) | 0.617 |
| Echocardiographic | | | | | |
| LVEDD (mm) | 43.0 (40.0–47.0) | 43.0 (40.0–47.0) | 44.0 (40.0–47.0) | 42.0 (39.0–45.0) | 0.120 |
| LA diameter (mm) | 40.0 (36.0–45.0) | 40.0 (35.0–45.0) | 41.5 (37.0–46.0) | 39.0 (35.0–44.0) | 0.023* |
| MWT (mm) | 19.0 (17.0–22.0) | 19.0 (17.0–21.0) | 19.0 (17.0–22.0) | 19.0 (17.0–22.0) | 0.626 |

**Table 1** (*continued*)

| Variable | All patients (*n* = 354) | LMR, tertile 1 (*n* = 115) | LMR, tertile 2 (*n* = 116) | LMR, tertile 3 (*n* = 123) | *p* value |
|---|---|---|---|---|---|
| LVEF (%) | 69.0 (63.0–73.0) | 69.0 (63.0–73.0) | 68.0 (63.0–71.0) | 70.0 (64.0–73.0) | 0.044* |
| Resting LVOTG ≥ 30 mm Hg | 147 (41.5%) | 47 (40.9%) | 51 (44.0%) | 49 (39.8%) | 0.799 |

**Notes.**

Values are mean ± SD or median (IQR) or n (%). LMR tertiles 1 to 3 were defined by <3.8, 3.8 to 5.4, and ≥5.5, respectively.

HCM, hypertrophic cardiomyopathy; SCD, sudden cardiac death; TE, thrombo-embolic event; ACE, angiotensin-converting enzyme; ARB, angiotensin receptor blocker; ICD, implantable cardioverter defibrillator; WBCC, white blood cell count; LMR, lymphocyte to monocyte ratio; LDL-C, low density lipoprotein cholesterol; HDL-C, high density lipoprotein cholesterol; TG, triglyceride; LVEDD, left ventricular end-diastolic dimension; LA, left atrial; MWT, maximal LV wall thickness; LVEF, left ventricular ejection fraction; LVOTG, left ventricular outflow tract gradient.

*Significant *p* values.

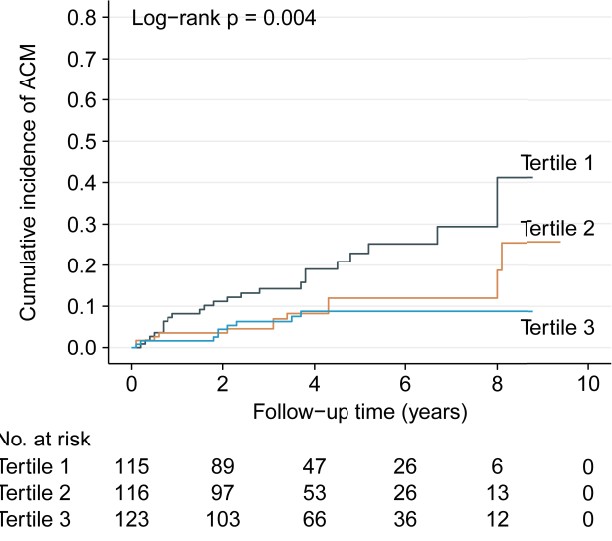

**Figure 1** **Kaplan–Meier analysis showing cumulative all-cause mortality by baseline lymphocyte to monocyte ratio tertiles.** ACM, all-cause mortality.

9 non-cardiac deaths. The cause of death could not be confirmed in 4 patients. The cumulative incidence of all-cause mortality was 24.8% for the whole cohort.

Based on the baseline LMR, there were 23 all-cause mortality in the first terile, 12 in the second tertile and 9 in the third tertile. The corresponding mortality rates were 7.3 (95% CI [4.4–10.1]), 2.2 (95% CI [1.0–3.5]) and 1.0 (95% CI [0.3–1.6]) per 100 person years, respectively. Kaplan–Meier analysis also demonstrated gradually decreased cumulative incidence of all-cause mortality across the three tertiles with log-rank *p* = 0.004 (Fig. 1).

## Relation of LMR to all-cause mortality

Univariate Cox regression analysis revealed that dyspnea, atrial fibrillation, age and left atria diameter were significant risk factors of all-cause mortality, while hemoglobin, lymphocyte, triglycerides, low-density lipoprotein cholesterol were significant protective factors (Table 2). In comparison of the top tertile versus the bottom tertile of LMR, the crude hazard ratio (HR) of all-cause mortality was 0.32 (95% CI [0.15–0.69]; *p* = 0.004). Two models were constructed to examine the effect of comorbidities, laboratory parameters,

**Table 2  Univariate Cox regression analysis of all-cause mortality.**

| Variable | Change | HR (95% CI) | p value |
| --- | --- | --- | --- |
| Gender | Female vs. Male | 1.30 (0.72–2.35) | 0.388 |
| Family history of HCM | Yes vs. no | 0.82 (0.30–2.31) | 0.713 |
| Family history of SCD | Yes vs. no | 2.00 (0.72–5.61) | 0.186 |
| Chest pain | Yes vs. no | 0.73 (0.40–1.32) | 0.300 |
| Palpitation | Yes vs. no | 0.74 (0.40–1.37) | 0.334 |
| Syncope/pre-syncope | Yes vs. no | 0.69 (0.36–1.34) | 0.277 |
| Dyspnea | Yes vs. no | 2.51 (1.27–4.97) | 0.008[*] |
| Hypertension | Yes vs. no | 1.02 (0.54–1.93) | 0.943 |
| Diabetes | Yes vs. no | 1.36 (0.54–3.46) | 0.515 |
| Prior TE | Yes vs. no | 2.19 (0.78–6.14) | 0.134 |
| Vascular disease | Yes vs. no | 1.38 (0.49–3.87) | 0.544 |
| Atrial fibrillation | Yes vs. no | 2.57 (1.38–4.79) | 0.003[*] |
| Aspirin | Yes vs. no | 1.15 (0.55–2.39) | 0.714 |
| Clopidogrel | Yes vs. no | 0.45 (0.06–3.29) | 0.434 |
| $\beta$-blocker | Yes vs. no | 0.62 (0.33–1.16) | 0.138 |
| ACE inhibitor or ARB | Yes vs. no | 0.57 (0.23–1.45) | 0.240 |
| Procedures | | | |
|   Alcohol septal ablation | Alcohol septal ablation vs. none | 0.50 (0.12–2.06) | 0.335 |
|   Septal myectomy | Septal myectomy vs. none | 1.63 (0.22–11.9) | 0.631 |
| Devices | | | |
|   Pacemaker | Pacemaker vs. none | 1.60 (0.49-5.19) | 0.431 |
|   ICD | ICD vs. none | 0.50 (0.12–2.09) | 0.345 |
| Resting LVOTG $\geq$ 30 mm Hg | Yes vs. no | 1.38 (0.76–2.50) | 0.283 |
| Age (years) | Per 1-SD increase | 1.51 (1.09–2.11) | 0.015[*] |
| Hemoglobin (g/L) | Per 1-SD increase | 0.67 (0.51–0.88) | 0.005[*] |
| WBCC ($10^9$/L) | Per 1-SD increase | 0.87 (0.62–1.23) | 0.439 |
| Neutrophil ($10^9$/L) | Per 1-SD increase | 1.04 (0.77–1.39) | 0.815 |
| Lymphocyte ($10^9$/L) | Per 1-SD increase | 0.61 (0.43–0.84) | 0.003[*] |
| Monocyte ($10^9$/L) | Per 1-SD increase | 1.12 (0.83–1.50) | 0.470 |
| TG (mmol/L) | Per 1-SD increase | 0.53 (0.32–0.89) | 0.016[*] |
| HDL-C (mmol/L) | Per 1-SD increase | 0.96 (0.71–1.30) | 0.803 |
| LDL-C (mmol/L) | Per 1-SD increase | 0.64 (0.47–0.88) | 0.006[*] |
| LVEDD (mm) | Per 1-SD increase | 0.85 (0.61–1.19) | 0.350 |
| LA diameter (mm) | Per 1-SD increase | 1.42 (1.08–1.86) | 0.012[*] |
| MWT (mm) | Per 1-SD increase | 1.03 (0.77–1.38) | 0.858 |
| LVEF (%) | Per 1-SD increase | 0.77 (0.60–0.99) | 0.044[*] |

**Notes.**
HR, hazard ratio; CI, confidence interval.
Other abbreviations as in Table 1.
*Significant p values.

**Table 3  Associations of LMR with all-cause mortality.**

| | LMR | | |
| --- | --- | --- | --- |
| | Tertile 1 | Tertile 2 | Tertile 3 |
| No. of patients (n) | 115 | 116 | 123 |
| Follow-up (PYs) | 317.2 | 534.7 | 926.5 |
| Deaths (n) | 23 | 12 | 9 |
| Mortality rates[a] (95% CI) | 7.3 (4.4–10.1) | 2.2 (1.0–3.5) | 1.0 (0.3–1.6) |
| Unadjusted HR (95% CI), p | Ref | 0.46 (0.23–0.93), 0.030[*] | 0.32 (0.15–0.69), 0.004[*] |
| Adjusted HR (95% CI), p | | | |
| Model 1 | Ref | 0.45 (0.22–0.90), 0.024[*] | 0.33 (0.15–0.72), 0.006[*] |
| Model 2 | Ref | 0.43 (0.20–0.91), 0.027[*] | 0.39 (0.17–0.90), 0.028[*] |

**Notes.**
[a] Per 100 PYs.
PYs,  person-years; CI,  confidence interval.
Other abbreviations as in Tables 1 and 2.
[*] Significant p values.
Model 1 with adjustment for age and sex.
Model 4 with adjustment for model 1 plus dyspnea, syncope/pre-syncope, family history of SCD, atrial fibrillation, hemoglobin, TG, LDL-C, LA diameter, LVEF, MWT and Resting LVOTG ≥30 mm Hg.

echocardiographic parameters and clinically relevant prognostic factors on the association between LMR and all-cause mortality risk. The association did not change materially after adjusting potential confounding factors. The fully adjusted HRs of all-cause mortality were 0.43 (95% CI [0.20–0.91], $p = 0.027$) for patients in tertile 2 versus tertile 1 and 0.39 (95% CI [0.17–0.90], $p = 0.028$) for patients in tertile 3 versus tertile 1, respectively (Table 3).

## Two-piecewise analysis and time-dependent AUCs
Due to the similar all-cause mortality rate in tertile 2 and tertile 3, we speculated that there might be a nonlinear association between LMR and all-cause mortality. Therefore, we further assessed the association between LMR and all-cause mortality by smooth curve fitting after adjustment for potential confounders, and a threshold effect was observed with a two-stage change and an inflection point. For LMR < 6.5, per SD increase resulted in a significantly decreased risk of all-cause mortality by 62% (adjusted HR: 0.38, 95% CI [0.21–0.68]), and the all-cause mortality risk did not progressively increase for the patients with a LMR ≥ 6.5 (Fig. 2A).

The time-dependent AUCs shown in Fig. 3A were developed on the basis of time-dependent ROC curves, which were calculated every four months. The AUCs indicated that LMR had a helpful discrimination for all-cause mortality in the whole cohort, with AUC ranged from 0.594 to 0.703 (Fig. 3A).

## Sensitivity analysis
Stratified analysis suggested the consistency of association between LMR and all-cause mortality remained in all subgroups, including age (<60 and ≥60 years), gender (male and female), AF (yes and no), beta-blocker (yes and no), hemoglobin (<139 and ≥139 g/L), WBCC (<6.3 and ≥6.3 $\times 10^9$/L), left atria diameter (<40 and ≥40 mm), left ventricular ejection faction (<50 and ≥50%) and left ventricular outflow tract gradient (yes and no), and there were no interactions between LMR and the above-mentioned variables.

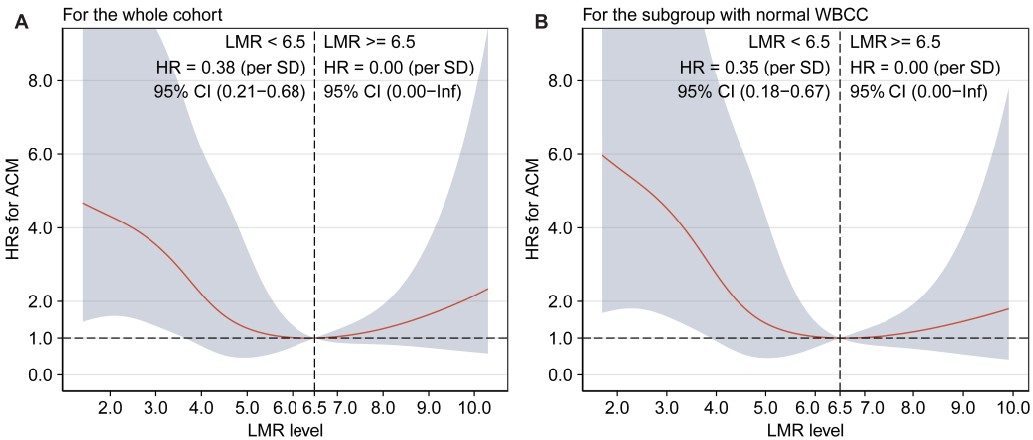

**Figure 2  Association between LMR and all-cause mortality by smooth curve fitting (A–B).** HR was adjusted for the covariates, which were the same as those in the model 2, and the red lines and blue ribbons depict the adjusted HRs and 95%CI. Abbreviations as in Tables 1 and 2.

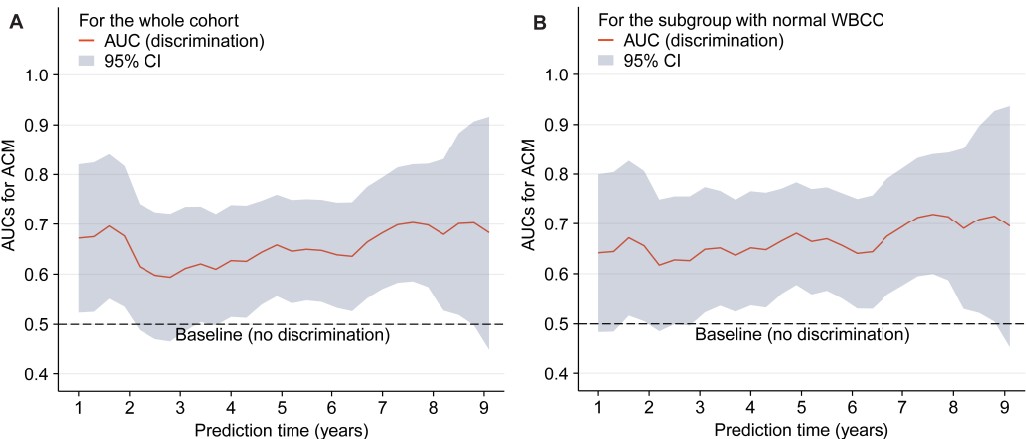

**Figure 3  Time-dependent AUCs (A–B).** The red lines and grey ribbons depict the AUCs and 95%CI. Abbreviations as in Tables 1 and 2.

Furthermore, we generated an *E*-value to assess the sensitivity to unmeasured confounding in the whole cohort. Based on the model 4 in the Table 3, the *E*-value was 2.97 for the tertile 2 of LMR, and the upper limit of the confidence interval was 1.34. For the tertile 3, the E-Value was HR = 3.22 (upper limit 1.36). For per SD increase of LMR (<6.5), the E-values for the point estimate and upper confidence bound for all-cause mortality were 3.29 and 1.94, respectively. Combined with the present multivariate results (Table S1) and the published data (*Park et al., 2018*), it is not likely that an unmeasured or unknown confounder would have a substantially greater effect on all-cause mortality than these known risk factors by having a HR exceeding about 3.0.

Finally, another sensitivity analysis including patients with normal WBCC ($4-10*10^9$/L) revealed similar results with the main analyses. Kaplan–Meier curve showed a gradually higher mortality risk with lower LMR (log-rank = 0.002). The adjusted HRs were significantly higher in the tertile 2 (HR = 0.39, 95% CI [0.18–0.86], $p = 0.019$), and in the tertile 3 (HR = 0.31, 95% CI [0.12–0.77], $p = 0.012$) compared with the tertile 1. In addition, adjusted HR of all-cause mortality was 0.35 (95% CI [0.18–0.67]) for per SD increase in the patients with a LMR <6.5, and the all-cause mortality risk did not significantly increase when LMR $\geq$ 6.5 (Fig. 2B). Figure 3B shows that LMR had a helpful discrimination for mortality, with AUC ranged from 0.617 to 0.720.

## DISCUSSION

In this study, we evaluated the value of LMR on admission in predicting the all-cause mortality in patients with HCM. To the best of our knowledge, this study appears to be the first to assess the predictive value of LMR for all-cause mortality in HCM patients. The results of the present study revealed that LMR was a novel prognostic indicator for all-cause mortality and that a higher LMR was significantly associated with better clinical outcomes. What's more, a non-linear relationship and a threshold effect with the inflection point at 6.5 was observed.

During the past decades, the mortality rate in HCM patients has been largely reduced with contemporary therapeutic strategies, which included implantable defibrillators to prevent sudden death, surgical myectomy or alcohol septal ablation to reverse obstructive heart failure, anticoagulation to reduce embolic stroke caused by atrial fibrillation and even heart transplantation to treat nonobstructive end-stage disease (*Maron, 2018*). With the use of such treatment interventions, the HCM related mortality rate can be as low as 0.5% per year (*Maron et al., 2015*; *Maron et al., 2016*). However, the situation is not satisfactory in developing country (*Zhu et al., 2017*; *Wang et al., 2020*). For example, in an observation study cohort consisted of 490 patients from Fuwai Hospital, China, *Zhu et al. (2017)* reported that the all-cause mortality rate and cardiovascular death rate were 2.12% and 1.67% per year, respectively. In this context, stratifying HCM patients for higher risk of adverse outcomes and the following close monitoring in clinical practice could potentially improve the disparities in the care of HCM patients across the world. According to the results of the present study, LMR, as an inexpensive and easily accessible marker, might afford a useful screening tool in HCM patients.

HCM has been ascribed to single sarcomere gene mutations. However, the heterogeneity of HCM clinical and pathobiological features could not solely be explained by a single molecular event (*Maron et al., 2019*). Other alternative and complementary mechanisms might be involved. Myocardial inflammation has recently been indicated as a contributor of cardiac hypertrophy, fibrosis and dysfunction among heart diseases (*Monda et al., 2020*), which are all recognized features of HCM. During the inflammatory process, immune cells played an important role. Neutrophils and monocytes would infiltrate in the injury site and release a bunch of harmful mediators to initiate the inflammatory response. Furthermore, lymphocytes take part in the sustention of inflammation via several effectors

(*Epelman, Liu & Mann, 2015*). However, the trigger for early inflammation in HCM has not been clearly defined yet. The intrinsic cardiomyocyte disarray, sarcomere injury and microvascular dysfunction seem to be related (*Monda et al., 2020*).

Cardiac inflammation leads to myocardial fibrosis (*Suthahar et al., 2017*; *Westermann, 2012*), which is a hallmark of HCM and regarded as a substrate for arrhythmias and heart failure (*Ho et al., 2010*). The worst complications of HCM mainly consisted of sudden cardiac death, heart failure related death and stroke related death. Both the life-threatening arrhythmias and progressively decompensated heart failure in HCM are significantly determined by myocardial fibrosis (*Bittencourt et al., 2019*; *O'Hanlon et al., 2010*). Therefore, the presence of cardiac inflammation and fibrosis might be the underlying mechanism of LMR as a prognostic factor in HCM patients. Unfortunately, there is a knowledge gap in terms of the relationship between the level of LMR and myocardial fibrosis in HCM patients and further prospective studies concerning this issue are warranted.

The greatest strength of our study is that this is the first study to illustrate the prognostic value of LMR in HCM patients. LMR act as a strong and independent predictor of all-cause mortality in patients with HCM. Since LMR is a cost-effective and widely accessible marker, it may help clinicians to identify high risk patients who require closer care and distribute the medical resource more efficiently. Besides, we have used the *E*-value sensitivity analysis to quantify the potential implications of unmeasured confounders and found that an unmeasured confounder was unlikely to negate the prognostic value of LMR. Several limitations also existed in the present study. First, those patients were from a single tertiary referral center, lack of region diversification and race comparison; thus, the generalization of our conclusion into other populations should be cautious. Second, this is a retrospective study, and data collection biases might potentially exist. Third, we only assessed the baseline LMR values in the prediction of all-cause mortality. The absence of serial LMR measurements during the follow-up time is one of the most handicaps. Fourth, the study failed to measure inflammatory cytokines, including C reactive protein, interleukin-6 and tumor necrosis factor, and these indicators may better help to explain the relation between LMR and poor outcomes. Fifth, Since the study endpoint were defined as all-cause mortality, the non-cardiac death and unknown death were also included for analysis. It might provide more useful information if only HCM-related mortality was analyzed. Due to the limited number of study endpoint, other large-scale studies based on multiple medical centers are encouraged.

## CONCLUSION

Our study denoted that LMR on admission was a novel and independent predictor for all-cause mortality in patients with HCM. This marker could be useful when applied for risk stratification of adverse clinical outcomes in HCM patients.

### Funding

This study was supported by the National Natural Science Foundation of China (Grant number: 81600299). The funders had no role in study design, data collection and analysis, decision to publish, or preparation of the manuscript.

### Grant Disclosures

The following grant information was disclosed by the authors:
National Natural Science Foundation of China: 81600299.

### Competing Interests

The authors declare there are no competing interests.

### Author Contributions

- Zhonglan Chen conceived and designed the experiments, performed the experiments, analyzed the data, authored or reviewed drafts of the paper, and approved the final draft.
- Ziqiong Wang conceived and designed the experiments, performed the experiments, analyzed the data, prepared figures and/or tables, authored or reviewed drafts of the paper, and approved the final draft.
- Youping Li performed the experiments, analyzed the data, authored or reviewed drafts of the paper, and approved the final draft.
- Xiaoping Chen performed the experiments, prepared figures and/or tables, authored or reviewed drafts of the paper, and approved the final draft.
- Sen He conceived and designed the experiments, authored or reviewed drafts of the paper, and approved the final draft.

### Human Ethics

The following information was supplied relating to ethical approvals (i.e., approving body and any reference numbers):

This study was approved by the Biomedical Research Ethics Committee, West China Hospital of Sichuan University (approval number: 2019-1147), and was conducted according to the criteria set by Declaration of Helsinki.

### Data Availability

The raw data is available in the Supplemental File.

### Supplemental Information

Supplemental information for this article can be found online at http://dx.doi.org/10.7717/peerj.13212#supplemental-information.

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
