# Peer review of "Relation between lymphocyte to monocyte ratio and survival in patients with hypertrophic cardiomyopathy: a retrospective cohort study"

_PeerJ, doi:10.7717/peerj.13212_

## Round 0.1 · original submission · Major Revisions

In the introduction, the authors may add a few sentences to a) describe further the relationship between hypertrophic cardiomyopathy (HCM) and lymphocyte to monocyte ratio (LMR), and b) some recent biomarkers.

The reviewers have raised a few issues in the methodology section. We suggest the authors provide more detailed information on the study, especially the data collection, sampling and sample size. Also, add a few references to support the methods chosen.

Be clear on the presentation of the results. Add necessary parameters to the tables and interpret the tables and figures as straightforward as possible.

Reviewer 1 ·

Basic reporting

Generally clearly written article with good language and overall structure.
Abstract
- generally ok

Introduction
- a bit too short. Maybe can highlight why LMR is considered better than current biomarkers or for complementary purposes.

Methods
- study design - not mentioned
- to detail out the sampling method and technique. 499 is universally sampled, any random sampling?
- when was the enrollment started and completed endpoint
- ethics - as informed consent was waived, what other steps were taken to ensure confidentiality and safeguard the data
- stats analysis - how and why did the author decide to split into 3 tertiles. reason to split into 3 and how they decided on the range and cut off points.

Results
- can elaborate further on the non-cardiac deaths and why they should be included in the study
- Tables- maybe can add * for the significant p-value/ values

References
- very few from 2020 reference, none from 2021. if possible to add more, but understandable if it is rare to find studies related to this topic.
-

Experimental design

No comment

Validity of the findings

Study has meaningful outcome and benefits.
Tertile was divided into 3 with stated range. However, it would be beneficial to add reference or source of the cut off points for the ranges.

Reviewer 2 ·

Basic reporting

The writing of the manuscript was clear and easy to understand.

Experimental design

1. Introduction: The justification of chosen LMR was not clearly stated. The authors need the explain how and why the decision was made to choose LMR as a prognostic factor.
2. Methods: Please add on the inception cohort as the study design was a retrospective cohort. The authors need to state the starting and endpoint clearly (What is the starting point? when is the endpoint, especially for censored cases especially the authors, mention on the end of follow up..when?).
Please calculate the minimum sample size needed for this study.
What sampling methods do the authors use to select the respondents?
Line 93-95: On what basic this LMR was divided into 3? Any citation?
What is the reason choose a time-dependent receiver operating characteristic (ROC) curve?

Validity of the findings

The manuscript explained clearly the statistical analysis and result with various analyses to control all the confounders.
The conclusion is linked with the research objective of the study.

Additional comments

No comments

Reviewer 3 ·

Basic reporting

Thank you for the opportunity to review this manuscript. It’s an interesting study, however, there are a few issues that need to be cleared. In the introduction, its more beneficial if the author could elaborate more on the relationship between hypertrophic cardiomyopathy (HCM) and lymphocyte to monocyte ratio (LMR). How the lymphocyte to monocyte ratio could affect the hypertrophic cardiomyopathy

Experimental design

For the methodolody – is there any reference age included in this study? Sample size calculation and sampling method were not mention in the methodology. LMR was divided into 3 tertiles; however, there was no reference cited in the methodology. Kindly mentioned in the method, what is the accrual time, the additional follow uptime, and when is the closure of the study

Validity of the findings

Overall, results were presented in tables and figures. However, few results need explanation. Line 124 - author mentioned WBCC increased across LMR tertiles, however, the result in table 1 showed the contrary. Figure 1- Kaplan Meier demonstrated a decreased cumulative incidence of all-cause mortality? However the figure showed the contrary. Table 2 is intended for univariate cox regression analysis of all-cause mortality- is it without considering the tertile? Why is tertile not included in the univariate analysis? Table 3, what type of variable is included in model 2. Table 3: adjusted HR for LMR for tertile 2 and 3 were 0.43 and 0.39 respectively – which showed LMR had a protective effect for all-cause of mortality- In the discussion –(Line 193) – lower LMR was significantly associated with adverse outcomes? Kindy explains.

---

## Round 0.2 · Minor Revisions

There are only small edits necessary (see the pdf file attached).

Reviewer 1 ·

Basic reporting

Authors have made changes according to comments.
Overall well written.

Experimental design

No comment

Validity of the findings

no comment

Reviewer 2 ·

Basic reporting

The authors has improved it.

Experimental design

The explanation and correction was done according to the comments

Validity of the findings

Well explained

---

## Round 0.3 · accepted · Accept

The authors have addressed all the comments and the article is now much improved as compared to the first version.